# Ag-Activated Metal−Organic Framework with Peroxidase-like Activity Synergistic Ag^+^ Release for Safe Bacterial Eradication and Wound Healing

**DOI:** 10.3390/nano12224058

**Published:** 2022-11-17

**Authors:** Jie Zhou, Ning Chen, Jing Liao, Gan Tian, Linqiang Mei, Guoping Yang, Qiang Wang, Wenyan Yin

**Affiliations:** 1Jiangxi Key Laboratory for Mass Spectrometry and Instrumentation, Jiangxi Province Key Laboratory of Synthetic Chemistry, East China University of Technology, Nanchang 330013, China; 2CAS Key Laboratory for Biomedical Effects of Nanomaterials and Nanosafety, Institute of High Energy Physics, Chinese Academy of Sciences, Beijing 100049, China; 3Laboratory for Micro-Sized Functional Materials, Department of Chemistry and College of Elementary Education, Capital Normal University, Beijing 100048, China; 4School of Energy and Chemical Engineering, Ulsan National Institute of Science and Technology (UNIST), Ulsan 44919, Republic of Korea; 5Institute of Pathology, The First Affiliated Hospital, Third Military Medical University, Chongqing 400038, China

**Keywords:** antibacterial, silver ions release, nanozyme, metal−organic framework, wound healing

## Abstract

Silver nanoparticles (Ag NPs), a commonly used antibacterial nanomaterial, exhibit broad-spectrum antibacterial activity to combat drug-resistant bacteria. However, the Ag NPs often causes a low availability and high toxicity to living bodies due to their easy aggregation and uncontrolled release of Ag^+^ in the bacterial microenvironment. Here, we report a porous metal−organic framework (MOF)-based Zr-2-amin-1,4-NH_2_-benzenedicarboxylate@Ag (denoted as UiO-66-NH_2_-Ag) nanocomposite using an in-situ immobilization strategy where Ag NPs were fixed on the UiO-66-NH_2_ for improving the dispersion and utilization of Ag NPs. As a result, the reduced use dose of Ag NPs largely improves the biosafety of the UiO-66-NH_2_-Ag. Meanwhile, after activation by the Ag NPs, the UiO-66-NH_2_-Ag can act as nanozyme with high peroxidase (POD)-like activity to efficiently catalyze the decomposition of H_2_O_2_ to extremely toxic hydroxyl radicals (·OH) in the bacterial microenvironment. Simultaneously, the high POD-like activity synergies with the controllable Ag^+^ release leads to enhanced reactive oxygen species (ROS) generation, facilitating the death of resistant bacteria. This synergistic antibacterial strategy enables the low concentration (12 μg/mL) of UiO-66-NH_2_-Ag to achieve highly efficient inactivation of ampicillin-resistant Escherichia coli (Amp^r^
*E. coli*) and endospore-forming Bacillus subtilis (*B. subtilis*). In vivo results illustrate that the UiO-66-NH_2_-Ag nanozyme has a safe and accelerated bacteria-infected wound healing.

## 1. Introduction

Nowadays, bacterial infection-induced diseases, such as skin abscesses [1,2], respiratory tract infections [3], and urinary tract infections, have become one of the most serious public problems in the world [4,5]. In the past few decades, many strategies have been made to fight against bacterial infections, and a range of antibiotics have been developed [6,7]. However, owing to the abuse of antibiotics, many bacteria emerged multi-drug resistance (MDR), resulting in increasingly difficult antibiotic treatments [8,9,10,11]. Therefore, it is very urgent to develop innovative antimicrobial systems and therapies as a complement to current antimicrobial treatments [12,13]. 

Nanomaterials have attracted great attention in antibacterial therapy due to their unique physicochemical properties, allowing them to have special therapeutic approaches to avoid the MDR of antibiotics. For examples, a large number of novel antibacterial nanomaterials such as precious metals [14], nano-metal oxides [15,16], metal-organic frameworks (MOFs) [17,18], organic nano-polymers [19,20], and carbon-based nanomaterials [21,22] have been developed. Typical antibacterial therapies such as enzyme-like catalysis therapy [23], photothermal therapy [24], photodynamic therapy [25], gas therapy [26], and combination therapy have been rationally designed. Among them, silver nanoparticles (Ag NPs) have broad-spectrum antimicrobial activity and can decrease MDR [27,28,29]. Moreover, Ag NPs have been used in clinical antibacterial applications [30]. Although their antibacterial mechanisms remain controversial, there is growing evidence that released Ag^+^ from Ag NPs are involved in key inhibition processes through membrane/wall deformation and production of reactive oxygen species (ROS). Consequently, the Ag NPs inhibit bacterial cell metabolism and eventually lead to bacteria death [31,32,33,34,35,36]. However, clinical studies have found that overuse of difficult-to-handle Ag NPs have frequently led to toxicity in humans, such as gastrointestinal disorders, convulsions, and even death [37]. In addition, increased Ag concentration does not necessarily lead to increased Ag^+^ concentration in the biological system. If Ag^+^ cannot be controllably released from the Ag NPs, amplified toxic effect of containing-Ag nanomaterials will occur [38,39]. Particularly, the Ag NPs alone or its composites with other organic polymers tend to aggregate due to poor compatibility and inhomogeneous distribution when exposed to the bacterial microenvironment, resulting in the reduction or elimination of antibacterial activity [40]. Therefore, it is a great challenge to explore novel antibacterial strategies to achieve efficient antibacterial activity using a smart nanocarrier that can carry Ag NPs homogeneously and avoid aggregation induced by direct contact between Ag NPs and the bacterial environment, ensuring sufficient use and controllable release of Ag^+^.

MOFs are novel multifunctional precursors composed of metal nodes and organic connections. Due to their unique advantages, such as porous structure and high specific surface area [41,42,43], the applications of MOFs have been extended to various biomedical fields, including phototherapy and nanocarriers, etc. [44,45]. It has been reported that the diversity of metal nodes, linkers and their wide range of coordination interactions can endow MOFs and their derivatives with enzyme-like catalytic activities, that is, nanozymes [46]. These MOFs-based nanozymes have gained increased attention owing to their low cost, durability, and high stability. Recently, the continuous development of MOFs-based nanozymes with peroxidase (POD)-like catalytic activity showed great potential in fighting against bacterial infection. This POD-like activity can decompose low dose H_2_O_2_ to generate highly toxic hydroxyl radicals (·OH), and then cause lipid peroxidation of bacteria [47]. Copper, iron, and zinc-based MOFs have been used to load Ag NPs in their frames [48,49]. However, unlike the biocompatible Zr-based nanomaterials [50], these MOFs have potential health hazards due to the metals used in their formations. Zr-2-amino-1,4-benzenedicarboxylate (UiO-66-NH_2_) is a typical Zr-based MOF, low-toxic and easily formed with good thermal and chemical stability, making it a suitable drug delivery carrier [51]. In addition, some studies have shown that exogenous metal can regulate the POD-like activity of UiO-66-NH_2_ nanozymes [52]. Therefore, we anticipate that the UiO-66-NH_2_ could be an ideal carrier for Ag NPs loading with good dispersion to form a smart enzyme-like catalytic platform. 

In this study, we designed a MOF-Ag-derived Zr-2-amino-1,4-benzenedicarboxylate@Ag (denoted as UiO-66-NH_2_-Ag) nanocomposite as a typical nanozyme with enhanced POD-like activity synergistic pH-responsive Ag^+^ release to achieve high availability and bactericidal effect. The anchoring of a small amount of Ag NPs on the porous surface of UiO-66-NH_2_ was successfully achieved via an in-situ immobilization method (Figure 1). Moreover, the porous surface of UiO-66-NH_2_ promoted the formation of ultra-small and well-dispersed Ag NPs to avoid direct interaction with external environment. Consequently, the UiO-66-NH_2_-Ag reduced the use dose of Ag and ensured its own biosafety and controllability due to its pH-responsive Ag^+^ release in the acidic bacterial microenvironment. In addition, anchoring of Ag NPs greatly activated POD-like activity of the UiO-66-NH_2_, enabling it to catalyze the decomposition of H_2_O_2_ in bacteria into ·OH with synergistic Ag^+^ release to produce more ROS, greatly enhancing antibacterial effect. Systematic antibacterial experiments showed that the UiO-66-NH_2_-Ag at very low concentration (12 μg/mL) had a >99.99% bactericidal activity against Gram-negative ampicillin-resistant *Escherichia coli* (Amp^r^ *E. coli*) and Gram-positive endospore-forming *Bacillus subtilis* (*B. subtilis*). Furthermore, their efficient antimicrobial effect and accelerated wound healing ability were successfully demonstrated in Amp^r^ *E. coli* infected skin of mice. Overall, the UiO-66-NH_2_-Ag nanocomposite possesses great potential for broad-spectrum bactericidal application.

## 2. Materials and Methods

**Materials.** Silver nitrate (AgNO_3_, 99.8%), zirconium tetrachloride (ZrCl_4_, >99.5%), 2-aminoterephthalic acid (ATA, >98%) and N,N-dimethyl formamide (DMF, >99.8%) were obtained from Aladdin. Disodium terephthalate (TA) was purchased from Alfa Aesar Reagent Co. 3,3′,5,5′-Tetramethylbenzidine (TMB) was obtained from Tokyo Chemical Industry. Nitric acid (HNO_3_, 70%, BV-III grade), hydrogen peroxide (H_2_O_2_, 30%), sodium chloride (NaCl), sodium hydroxide (NaOH), ethanol, and anhydrous methanol (>99.7%) were obtained from Beijing Chemical Corporation, Beijing, China. Fetal bovine serum (FBS), yeast extract, dulbecco’s modified eagle’s medium (DMEM), penicillin, streptomycin, and tryptone were obtained by Sigma-Aldrich. SYTO 9 green fluorescent nucleic acid was purchased from Keygen Biotechnology. Hoechst 33342, 2′,7′-dichlorodihydrofluorescein diacetate (DCFH-DA) and propidium iodide (PI) were obtained from Beyotime, Shanghai, China. 

**Synthesis of UiO-66-NH_2_ nanoparticles.** UiO-66-NH_2_ nanoparticles (NPs) were synthesized via a hydrothermal approach in terms of an improved process [53]. In a typical synthesis, 30.3 mg (0.13 mM) ZrCl_4_ and 23.5 mg (0.13 mM) ATA were dissolved in DMF (15 mL) with stirring. Then, the above mixture was shifted to a sealed Teflon-lined stainless-steel autoclave (25 mL), heat at 200 °C for 1 d. After cooling, the resulting product was washed by centrifugation with anhydrous methanol. The final products were obtained after drying.

**Synthesis of UiO-66-NH_2_-Ag nanocomposite.** In-situ immobilization strategy was adopted to synthesize the UiO-66-NH_2_-Ag nanocomposite. First, the UiO-66-NH_2_ NPs (20 mg) were dispersed in anhydrous methanol (5 mL) under stirring for 20 min. Then, anhydrous methanol solution (5 mL) containing AgNO_3_ (10 mg) was added under stirring for another 20 min. The suspension was vigorously stirred in the dark under 50 °C. 20 h later, the obtained product UiO-66-NH_2_-Ag was centrifuged and washed several times with methanol and distilled water, and then dried at room temperature under dark.

**Characterizations.** A Bruker D8 (Germany) Advanced Power diffractometer (λ = 1.5406 A) was used for powder X-ray diffraction (XRD) analysis of samples. A Hitachi S-4800 (Tokyo, Japan) field emission scanning electron microscope (FE-SEM) and a Tecnai G2 F20 (Eindhoven, The Netherlands) transmission electron microscope (TEM) were used to characterize the morphology and size of samples. Element analysis result was obtained via an energy-dispersive X-ray spectrum (EDX, Oxford x-met 8000, Oxford, UK) attached on the TEM. A JEM-ARM 200F cold field gun (Tokyo, Japan, 200 kV) was used for high-angle annular dark-field scanning transmission electron microscope (HAADF-STEM) to analyze the structure and composition of samples. A Micromeritics Tristar 3000 (Norcross, GA, USA) instrument was used to analyze the pore size distributions and nitrogen adsorption-desorption isotherms at 77 K. The specific surface was calculated based on the relative pressure (P/P0) from 0.01 to 1.00 via the Brunauer-Emmett-Teller (BET) method. Escalab 250 X-ray photoelectron spectrometer (Thermo Fisher Scientific Inc., Waltham, UK) with Al K_α_ radiation was used for X-ray photoelectron spectroscopy (XPS) measurements. UV-Vis-NIR spectrophotometer (VARIAN CARY 50, Shanghai, China) was used to test the absorption peak. A Zeta potential/Particle system (Malvern Zetasizer Nano ZS, Worcestershire, UK) was used to measure Zeta potential.

**Analysis of pH-dependent Ag^+^ release from UiO-66-NH_2_-Ag nanocomposite.** UiO-66-NH_2_-Ag nanocomposite (40 mg) was dispersed at different pH values in phosphate buffered saline (PBS, pH 7.4 or 5.0, 20 mL) at 37 °C. Aliquots of solution (1 mL) were withdrawn at planned time points (45 min, 1.5, 3, 6, 12 h and 1, 2, 3, 4, 5, 6, 7, 10 d). After filtering the samples, the suspension solutions were shifted to flasks containing HNO_3_ (5 mL, 70%), and digested at 180 °C. Afterwards, inductively coupled plasma mass spectrometry (ICP-MS, Thermo Elemental X7, Winsford, UK) was used to detect the digested solutions. The detected samples were repeated three times.

**Detection of ·OH.** Non-fluorescent terephthalic acid (TA) can be oxidized by ·OH to 2-hydroxyl terephthalic acid (TAOH), which has a characteristic peak at 435 nm under the excitation of 315 nm [54]. By observing the fluorescence intensity of TAOH, we detected the formation of ·OH from H_2_O_2_ catalyzed by UiO-66-NH_2_-Ag using the Horiba FluoroLog-3 fluorescence spectrometer (USA). Typically, the solutions to be measured were divided into five groups, including TA, UiO-66-NH_2_-Ag, H_2_O_2_, TA + H_2_O_2_ and TA + H_2_O_2_ + UiO-66-NH_2_-Ag. The final working concentrations were 10 μg/mL for UiO-66-NH_2_-Ag, 0.1 mM for H_2_O_2_ and 0.5 mM for TA. The solution was gently mixed in the dark at 37 °C for 12 h. Then, the fluorescence intensity was recorded.

**POD-like catalytic activity of UiO-66-NH_2_-Ag.** TMB was used as a substrate to measure the POD-like activity of UiO-66-NH_2_-Ag. UV-Vis absorbance of oxidized TMB (oxTMB) at 652 nm will be observed using UV-Vis-NIR spectrophotometer (VARIAN CARY 50, Shanghai, China) if the UiO-66-NH_2_-Ag has an obvious POD-like activity, like other POD-like catalytic nanomaterials [55]. Typically, the TMB solution was added to HAc-NaAc buffer containing UiO-66-NH_2_-Ag (33 μg/mL) and H_2_O_2_ (10 mM) and incubated for 10 min at 25 °C in the dark to evaluate the POD-like activity by recording the absorption at 652 nm. The effect of H_2_O_2_ and UiO-66-NH_2_-Ag concentrations on POD-like activity was investigated by changing the concentrations of UiO-66-NH_2_-Ag (5, 10, 20, 50, 100 μg/mL) and H_2_O_2_ (0, 2.5, 5, 10, 20 mM) at 25 °C. In addition, the POD-like activity was also measured by varying the incubation temperature (25, 30, 35, 40, 50, 60, 70 °C) in the HAc-NaAc buffer and pH (2, 3, 4, 5, 6, 7, 8) in the PBS buffer. 

**Cytotoxicity and hemolysis analysis.** Human umbilical vein endothelial cells lines (HUVEC) and human glioma cells lines (U87) were employed for the investigation of the cytotoxicity of UiO-66-NH_2_-Ag. The two cells (6 × 10^3^ cells per well) were grown in 96-well plates and incubated in an incubator (37 °C, 5% CO_2_) for 24 h. Subsequently, the two cells were incubated with different concentrations of UiO-66-NH_2_-Ag (0, 3.9, 7.8, 15.6, 31.2, 62.5, 125, and 250 μg/mL) for a further 24 h. After being washed three times with fresh medium, Cell Counting Kit-8 (CCK-8) assay was adopted and then a microplate reader (SpectraMax M2, USA) was used to record the absorbance at 450 nm. 

For hemolysis analysis, fresh blood (1 mL) obtained from BALB/c mice (6 weeks old) obtained from Vital River was mixed with PBS (2 mL) containing EDTA. After centrifugation at 2000 rpm and washing three times, red blood cells (RBCs) were collected and resuspended in PBS. The diluted RBCs suspension solution was then added to PBS (negative control), distilled water (positive control) and different concentrations of UiO-66-NH_2_-Ag dispersions (3.9, 7.8, 15.6, 31.3, 62.5, 125, and 250 µg/mL). These mixtures were kept at 25 °C for 4 h. Finally, UV-Vis spectrophotometry was used to measure the absorbance of the supernatant at 541 nm.

**Bacterial solutions.** Gram-negative ampicillin-resistant *Escherichia coli* (Amp^r^ *E. coli*) and Gram-positive endospore-forming *Bacillus subtilis* (*B. subtilis*) were respectively transferred to Luria-Bertani (LB) containing ampicillin (50 µg/mL) and Beef-Peptone-Yeast (BPY) broth. The two bacteria were kept at 37 °C under shaking for 5 h. Then, the bacteria were centrifuged and washed with PBS. 

**In vitro antibacterial performances.** The antibacterial performances of UiO-66-NH_2_-Ag nanocomposite were tested by the plate counting method. Two Gram bacteria (Amp^r^ *E. coli* and *B. subtilis*, OD_600_ = 0.10) were assayed diluted to 1.0 × 10^5^ CFU/mL in PBS. The diluted bacteria (0.4 mL) were mixed with the UiO-66-NH_2_-Ag (0.1 mL) with the final concentration of 0, 3, 6, 12, 25, 50, and 100 μg/mL and oscillated at 37 °C for 4 h. Next, the above mixture (100 μL) was spread on LB or BPY solid culture plates and cultured at 37 °C for 24 h. 

Moreover, we also performed a live/dead staining experiment to study the antibacterial performance of UiO-66-NH_2_-Ag. First, UiO-66-NH_2_-Ag nanocomposite (0, 3, 6, 12 μg/mL) were incubated with 500 μL of Amp^r^ *E. coli* or *B. subtilis* suspension (1.0 × 10^8^ CFU/mL) at 37 °C for 4 h. After incubation, propidium iodide (PI) (red) and SYTO 9 (green) were mixed with the suspension of Amp^r^ *E. coli* or *Bacillus subtilis*. The final concentrations of PI and SYTO 9 were 30 μM and 20 μM, respectively. The mixture was incubated at 37 °C under dark for 25 min. Finally, laser scanning confocal microscopy (Nikon, Tokyo, Japan) was used to observe the fluorescence images.

**Morphology observation of bacteria.** To further evaluate antibacterial performance, Amp^r^
*E. coli* and *B. subtilis* were mixed with different concentrations (6, 12, 25, 50 and 100 μg/mL) of UiO-66-NH_2_-Ag nanocomposite at 37 °C for 4 h. Next, these bacteria were fixed with paraformaldehyde (4%) at 4 °C. Then, the bacteria were sequentially dehydrated with ethanol (30%, 50%, 70%, 80%, 90%, and 100%) for 10 min. The prepared bacteria were sprayed with gold and used for FE-SEM images.

**ROS detection.** DCFH-DA fluorescent probe was used to detect ROS production in bacteria. First, UiO-66-NH_2_-Ag nanocomposite (0, 3, 6, 12 μg/mL) were incubated with 500 μL of Amp^r^ *E. coli* or *B. subtilis* suspension (1.0 × 10^8^ CFU/mL) for 4 h at 37 °C. After incubation, DCFH-DA was mixed with suspensions of *E. coli* or *B. subtilis*. The final concentration of DCFH-DA was 10 μM. The mixture was incubated at 37 °C for 15 min under dark. Finally, the fluorescence images were observed using a laser scanning confocal microscope.

**In vivo wound healing.** To evaluate the performance of UiO-66-NH_2_-Ag on promoting wound healing, male BALB/c mice (6 weeks) were grouped into three groups (*n* = 6 in each group): (1) Control PBS; (2) Ag^+^; (3) UiO-66-NH_2_-Ag. After anesthetizing, the wound with diameter of 5 mm (∼78 mm^2^) was surgically obtained on the back of the mice. These skin wounds were then infected with Amp^r^ *E. coli* suspension (1.0 × 10^5^ CFU/mL). 12 h later, 5 μL of PBS, AgNO_3_ (0.86 μg/mL) and UiO-66-NH_2_-Ag (12.5 μg/mL with 0.86 μg of Ag loading) solutions were applied to the wounds at corresponding groups, respectively. This process of adding the above solutions to the wound was performed twice at 12-h intervals. The wounds were photographed daily, and the body weight of mice was recorded. During the treatment, wound tissues were photographed, and then the skin tissues (*n* = 3 per group) were collected and fixed with 4% formaldehyde for Hematoxylin-Eosin (H&E) staining and Masson’s staining within day 8. Furthermore, blood samples were collected from each group of mice on day 8 for routine blood analysis. After that, representative H&E-stained images of major organs of mice (*n* = 6 for each group) after being treated with UiO-66-NH_2_-Ag at day 7, day 15, and day 30 were also used to evaluate long-term safety. All animal experiments were performed and approved in accordance with the Key Laboratory for Biomedical Effects of Nanomaterials and Nanosafety, CAS guidelines for the care and use of laboratory of Animals Ethics Committee.

## 3. Results and Discussion

The porous MOF-based UiO-66-NH_2_ NPs were synthesized by a hydrothermal approach. Subsequently, an in-situ immobilization dispersion reaction of Ag^+^ and the UiO-66-NH_2_ was used to initiate and fix Ag NPs on the surface of UiO-66-NH_2_, resulting in the formation of MOF-based UiO-66-NH_2_-Ag nanocomposite. FE-SEM image in Appendix A and TEM image in Figure 1a showed that the UiO-66-NH_2_-Ag had an octahedral-like construction with a diameter of ~60 nm. Noteworthy, the Ag NPs with a diameter of about 5 nm were uniformly distributed on the surface of UiO-66-NH_2_. XRD patterns of the UiO-66-NH_2_ showed the purity and good crystallinity (Figure 1b) [56]. The high agreement between UiO-66-NH_2_ and UiO-66 on the 2θ peak indicated that the introduction of the -NH_2_ group had no impact on the skeletal structure of UiO-66. After in-situ immobilization of the Ag NPs, typical peaks (111 and 200) of cubic-phased Ag appeared (JCPDS No. 87-0719), implying the successful anchoring of Ag NPs on the surface of UiO-66-NH_2_. To evaluate the porosity of UiO-66-NH_2_, N_2_ adsorption/desorption isotherm was measured, and the specific surface area was calculated as 473 m^2^ g^−1^ (Appendix A). According to the Barrett–Joyner–Halenda (BJH) model, the pore size distribution from the isothermal adsorption branch was estimated to be approximately 1.5 nm (Appendix A). Furthermore, we found that UiO-66-NH_2_ had a low zeta potential in water (−5.5 ± 0.47 mV). However, the zeta potential of UiO-66-NH_2_-Ag changed to +8.06 ± 0.16 mV, further implying a successful immobilization of Ag to the UiO-66-NH_2_-Ag surface (Figure 1c).

The element mapping of Zr, N, Ag, and O in UiO-66-NH_2_-Ag nanocomposite was illustrated by HAADF-STEM images (Figure 1d). The EDX spectrum of UiO-66-NH_2_-Ag nanocomposite (Appendix A) exhibited a similar result with the elemental mapping analysis. Next, we analyzed the surface elemental compositions and chemical states of the obtained nanocomposite. As shown in Figure 1e, XPS survey spectrum indicated the coexistence of Zr, C, N, Ag and O in the UiO-66-NH_2_-Ag. Moreover, the XPS spectrum of Ag 3d was shown in Figure 1f, where the Ag 3d_5/2_ (368.5 eV) and Ag 3d_3/2_ (374.6 eV) peaks implied the existence of Ag in the zero-valent form [57]. The XPS Zr 3d peak could be further split into Zr 3d_5/2_ (182.6 eV) and Zr 3d_3/2_ (185.0 eV), which were attributed to Zr-O in UiO-66 (Appendix A) [58]. The N 1s XPS spectrum showed a typical peak located at 399.4 eV, corresponding to –NH_2_ and –NH groups (Appendix A). UV-Vis-NIR spectrum of the UiO-66-NH_2_-Ag well inherited the feature of UiO-66-NH_2_ and had wide absorbance from UV-Vis to NIR regions. Especially, UiO-66-NH_2_-Ag had a stronger absorbance than UiO-66-NH_2_ at 400–750 nm on account of the successful anchoring of Ag NPs (Figure 1g). 

Nanomaterials especially metal-based nanocomposites have the ability as nanozyme with improved POD-like catalytic activity, which can catalyze H_2_O_2_ to produce highly toxic ·OH for killing bacteria [16,17]. We then attempted to evaluate the POD-like activity of UiO-66-NH_2_-Ag nanocomposite by the catalytic oxidation of colorless TMB to blue oxidized TMB (oxTMB) in the presence of H_2_O_2_. In Figure 2a, compared with other groups, the TMB + H_2_O_2_ + UiO-66-NH_2_-Ag had an intense absorption at 652 nm with a significant blue color within 10 min, confirming that UiO-66-NH_2_-Ag had a high POD-like activity. Meanwhile, the blue gradually deepened with the increased UiO-66-NH_2_-Ag concentration, indicating the concentration-dependent POD-like activity (Figure 2b). The catalytic activity also depended on H_2_O_2_ concentration, pH values and temperature. As shown in Figure 2c, the POD-like activity increased with the increased H_2_O_2_ concentration. Moreover, we found that the optimal pH value and temperature were ~pH 5.0 and 37–42 °C, which were very close to the pH value of real bacterial microenvironment and living body temperature (Figure 2d,e). It is reasonable to refer that the enhanced POD-like activity of UiO-66-NH_2_-Ag after being activated by the Ag NPs could be due to the accelerated electron transfer from TMB to H_2_O_2_ compared with the UiO-66-NH_2_ NPs.

Due to the enhanced POD-like activity of UiO-66-NH_2_-Ag, it is necessary to further investigate the formation of ·OH. TA was used as a typical fluorescence (FL) probe to detect the ·OH with a maximum peak at 435 nm upon the reaction with ·OH. As shown in Appendix A, besides its inherent FL peak [59], the characteristic peak at 435 nm was significantly enhanced in TMB + H_2_O_2_ + UiO-66-NH_2_-Ag group compared to the TMB + UiO-66-NH_2_-Ag group, implying that H_2_O_2_ could be efficiently changed into ·OH catalyzed by UiO-66-NH_2_-Ag. The enhanced POD-like activity provided a possibility for the subsequent synergy with controllable Ag^+^ release to kill bacteria.

It was reported that the bacteria-infected acute wound microenvironment is weak acidity [60]. To investigate the pH-dependent Ag^+^ release capacity of UiO-66-NH_2_-Ag nanocomposite responsive to acute wound microenvironment, the Ag content on the surface of UiO-66-NH_2_-Ag was determined to be 4.396 wt% by ICP-MS. In addition, the cumulative Ag^+^ release capacity of UiO-66-NH_2_-Ag over 10 d is shown in Figure 2f. The release of Ag^+^ increased significantly within the first 2 h both in pH 7.4 and 5.0. Then, 47.33 µg/mL of Ag^+^ was released from UiO-66-NH_2_-Ag into 20 mL media within 1 d, and 56.12 µg/mL of cumulative release of Ag^+^ within 10 d was calculated. Especially, the cumulative Ag^+^ release capacity of UiO-66-NH_2_-Ag over 10 d can reach up to ~84.12 µg/mL, which was very higher than that in the neural condition. This process proves the pH-responsive release ability of UiO-66-NH_2_-Ag, which can be explained as follow. First, the porous UiO-66-NH_2_ scaffolds as a relatively stable nanocarrier make the Ag NPs well-dispersed in the biological environment and difficult to reunite. Second, the well-dispersed Ag NPs with large surface area fixed on the surface of UiO-66-NH_2_-Ag scaffolds could easily release the relatively large number of Ag^+^. At the same time, the positive surface charge of UiO-66-NH_2_-Ag (shown in Figure 1c) could also easily repel the released Ag^+^ to the surface of bacteria, and then more easily meets the release requirement into acute acidic wound microenvironment. The pH-responsive Ag^+^ release of the UiO-66-NH_2_-Ag offers great potential for chemical sterilization.

Inspired by the ability of controllable release of Ag^+^ and the remarkable POD-like activity, we next evaluated the antibacterial effects of UiO-66-NH_2_-Ag nanocomposite. Typical Gram-negative ampicillin-resistant *Escherichia coli* (Amp^r^ *E. coli*) was selected as a model bacterial strain, and the antibacterial ability was determined by plate counting method. Interestingly, the UiO-66-NH_2_-Ag had superior antibacterial efficacy and led to a very low survival rate of 2.2% for Amp^r^ *E. coli* after co-incubated with UiO-66-NH_2_-Ag (6 μg/mL) for 4 h (Figure 3a,c). However, ~98% survival rate of Amp^r^ *E. coli* was observed for UiO-66-NH_2_ without Ag NPs loading under the same incubation condition (Appendix A), suggesting the negligible antimicrobial effect of low concentration UiO-66-NH_2_. The large gap in antimicrobial effect of UiO-66-NH_2_ before and after loading of Ag NPs suggested the importance of the synergy of Ag^+^ release and the increased POD-like activity of UiO-66-NH_2_ after loading Ag NPs with uniform distribution. And the UiO-66-NH_2_-Ag can promote the production of a large amount of ROS with a concentration-dependent effectin bacterial microenvironment (Appendix A). To further investigate the antibacterial performance of UiO-66-NH_2_-Ag against other bacteria, endospore-forming *Bacillus subtilis* (*B. subtilis*) as a typical Gram-positive bacterium was selected for antibacterial test. As shown in Figure 3b,d, after 4 h of incubation, the bacterial survival rate decreased sharply with the increased concentration of UiO-66-NH_2_-Ag. Notably, the two bacteria died with >99.99% of killing rate when the concentration of UiO-66-NH_2_-Ag was 12 μg/mL. 

Next, live/dead staining was used to distinguish living cells from dead cells by SYTO 9 (for dead and live cells, green) and propidium iodide (PI for dead cells, red) staining for further understanding its concentration-dependent killing ability to Amp^r^ *E. coli* and *B. subtilis*. Similar to the previous plate counting results, more dead bacteria were observed in the 12 μg/mL of UiO-66-NH_2_-Ag group, while a few dead bacteria were observed when treated with 6 μg/mL and 3 μg/mL of UiO-66-NH_2_-Ag (Appendix A). The above results showed that this Ag-loaded nanocomposite exhibited remarkable antibacterial properties for Amp^r^ *E. coli* and *B. subtilis* at very low doses.

FE-SEM images in Figure 4a were used to further study the antibacterial behavior of UiO-66-NH_2_-Ag. As expected, the Amp^r^ *E. coli* incubated with control group was smooth. The rod-like morphology and the cell membrane structure of the Amp^r^ *E. coli* were intact. However, 6 μg/mL of UiO-66-NH_2_-Ag treated for 20 min resulted in slight wrinkles and some disruption on the surface of bacterial cell wall (red arrows). As the concentration of UiO-66-NH_2_-Ag rose to 12 μg/mL, the damage of bacterial surface was more intense with more wrinkles and incomplete surfaces. As the concentration increased to 50 μg/mL, the membrane of bacteria was severely damaged, and the bacterial morphology was changed and fused together. Similar results were verified on *B. subtilis* (Figure 4b). Therefore, UiO-66-NH_2_-Ag with low dose use of Ag NPs has a powerful antibacterial effect due to the high POD-like activity synergistic Ag^+^ release, which can produce large amounts of ROS to directly disrupt bacteria. This efficient antibacterial performance and low dose use of Ag NPs offer great possibilities for bacterial disinfection.

Good biocompatibility is a prerequisite for biological applications. Therefore, we performed hemolysis to assess the effect of UiO-66-NH_2_-Ag nanocomposite on RBCs. Interestingly, no obvious erythrocyte hemolysis was observed after incubation in RBCs with different concentrations of UiO-66-NH_2_-Ag (Appendix A). Furthermore, in vitro cytotoxicity studies of UiO-66-NH_2_-Ag nanocomposite on HUVEC and U87 cells were performed by CCK-8 assay. Even at 250 μg/mL of UiO-66-NH_2_-Ag, the two cells remained highly active (Appendix A). These results clearly revealed that UiO-66-NH_2_-Ag has good biocompatibility at the tested dosage. 

To observe the in vivo wound healing effect of UiO-66-NH_2_-Ag nanocomposite, Amp^r^ *E. coli* was selected to infect the wounds on the epidermis of BALB/c mice (Figure 5a). The mice were divided into (1) Control, (2) Ag^+^, and (3) UiO-66-NH_2_-Ag groups. Figure 5b showed representative photographs of different treated wounds in mice during the treatment. All wounds shrank with the prolonged time. Notably, the UiO-66-NH_2_-Ag treated group showed better wound healing than the control and Ag^+^ groups, which can be attributed to the effective Ag^+^ release synergetic POD-like activity on the wound surface. The maximum percentage of wound area in each group further demonstrated its effect in accelerating wound healing (Figure 5c). There was no obvious change in the body weight of mice for UiO-66-NH_2_-Ag treated group compared with the control group, indicating that UiO-66-NH_2_-Ag had no significant toxicity to mice (Figure 5d). After treatment, the wound tissues were analyzed by H&E staining and Masson trichrome staining (Figure 5e and Appendix A). On day 3, H&E staining showed epidermal damage in each group. On day 5, repair of damaged skin tissues was observed in all treatment groups. On day 8, complete reepithelialization and differentiated epithelium were clearly observed in the UiO-66-NH_2_-Ag group compared to the other groups. Masson trichrome staining (blue) was used to distinguish collagen fibers from muscle fibers during wound healing. On days 3 and 5, no obvious collagen fibers were observed in each group. On day 8, collagen fibers in control group had no obvious recovery effect, while collagen fibers in Ag^+^ group had some recovery. The UiO-66-NH_2_-Ag group showed the optimal recovery of collagen fiber. In addition, major organs of mice were stained with H&E and blood indexes of mice were used to evaluate the biocompatibility of the UiO-66-NH_2_-Ag. As shown in Appendix A, no obvious damage or inflammatory reaction was observed in the H&E staining, indicating that the UiO-66-NH_2_-Ag had good histocompatibility within 30 days. Meanwhile, the nine blood routine indexes were all within the normal range, indicating that UiO-66-NH_2_-Ag was biologically safe (Appendix A). Therefore, this synergistic antibacterial system can not only achieve rapid wound disinfection, but also has good biosafety.

## 4. Conclusions

In summary, we constructed a biocompatible MOF-based UiO-66-NH_2_-Ag nanocomposite consisting of well-dispersed Ag NPs anchored on the porous UiO-66-NH_2_. The nanocomposite can avoid the aggregation of Ag NPs while enhancing dispersion of Ag NPs. Meanwhile, UiO-66-NH_2_-Ag not only improved the utilization of Ag NPs, but also provided a pH-responsive Ag^+^ release in acidic wound environment. In addition, the UiO-66-NH_2_-Ag can act as nanozyme with enhanced POD-like catalytic activity after loading Ag NPs, which allowed it to break down H_2_O_2_ within bacteria to produce highly toxic ·OH to destroy bacteria integrity. This POD-like activity synergistic Ag^+^ release greatly facilitated the production of more ROS in bacteria, resulting in highly effective bactericidal activity. UiO-66-NH_2_-Ag at a low concentration of 12 μg/mL could effectively kill Amp^r^
*E. coli* and *B. subtilis* with >99.99% bactericidal activity. Moreover, UiO-66-NH_2_-Ag maintained low cytotoxicity over a wide concentration range (0–250 μg/mL). In vivo results revealed that UiO-66-NH_2_-Ag exhibited high antibacterial effect and significantly accelerated bacteria-infected wound healing without any influence on organ and blood indicators. This work provides a new strategy for safe wound disinfection broadens the prospect of antimicrobial nanomaterials for broad-spectrum bactericide application.

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
