# Peer review of "Ag-Activated Metal−Organic Framework with Peroxidase-like Activity Synergistic Ag+ Release for Safe Bacterial Eradication and Wound Healing"

_nanomaterials, 2022, doi:10.3390/nano12224058_

Round 1
Reviewer 1 Report
The topic discussed in the paper "Ag-activated metal−organic framework with peroxidase-like 2 activity synergistic Ag+ release for safe bacterial eradication and 3 wound healing" is of interest nowadays because of multidrug resistance issue.
However, the paper must be improved, mostly in the experimental part, because is very confused and confusing as it is.
Here below some detailed comments.
Introduction: I think the authors should explain the acronym UiO-66-NH2-Ag.
Introduction, line 56: The sentence is not grammatically correct. Please correct verb tenses and subject.
Introduction, line 75: MOF, Please explain the acronym.
Materials and Methods, line 117: In my opinion the authors should subdivide the Materials and methods section in subsections dedicated to the specific experimentals, i.e. UiO-66-NH2 synthesis, Biological characterization, etc....
Materials an Methods, line 162: Please indicate name and address of ICP-MS manufacturer.
Materials and Methods, line 171: Please indicate which instrument was used to record fluorescence.
Materials and Methods, line 189: Please indicate which instrument was used to measure POD activity.
Materials and Methods, line 235: Please indicate the amount of UiO-66-NH2-Ag applied to each wound.
Results, line 264: Zeta potential is -5.5; it would better to write that zeta potential was close to zero.
Results, line 266: Zeta potential is 8.06. Even this zeta potentail is quite "low", that means close to zero. Zeta potential that would interact with cells should be at least higher than 10 - 15 mV. Thus, in my opinion, these values of zeta potentials (both negative and positive) are not relevant to an interaction with cells. Rebuttal from the authors is required, also on the basis of data that can be found in the literature.
Reviewer 2 Report
The manuscript “Ag-activated metal−organic framework with peroxidase-like activity synergistic Ag+ release for safe bacterial eradication and wound healing” proposes a porous metal−organic framework (MOF)-based UiO-66-NH2-Ag nanocomposite for improving dispersion and utilization of Ag NPs. The idea behind the work is interesting and several analyses were carried out to validate the composite system; however, some revisions are required before the publication:
- Abstract. Define all acronyms;
- The last paragraph of the Introduction is too similar to the Abstract. Rewrite or remove results.
- Remove the adjective “simple” to the production method, since it is constituted of several steps. Compare the obtained results with the previous literature in order to highlight the relevance of the present findings; for this purpose, see for instance these works: Baldino et al., Journal of Chemical Technology and Biotechnology, 2019, 94(1), pp. 98–108; Cuadra et al., Applied Sciences, 2022, 12(10), 5023; Lange et al., Materials, 15(9), 3122; etc..
- Correct typos.
